# Towards an Early Clinical and Biological Resistance Detection in Dermatophytosis: About 2 Cases of *Trichophyton indotineae*

**DOI:** 10.3390/jof9070733

**Published:** 2023-07-07

**Authors:** Giuseppe Russo, Laurence Toutous Trellu, Lionel Fontao, Béatrice Ninet

**Affiliations:** 1Division of Dermatology and Venereology, University Hospital of Geneva, CH-1205 Geneva, Switzerland; 2Laboratory of Dermatology, Division of Laboratory Medicine, University Hospital of Geneva, CH-1205 Geneva, Switzerland

**Keywords:** *Trichophyton indotineae*, terbinafine-resistance, dermatophyte infection, tinea corporis, antifungal susceptibility

## Abstract

*Trichophyton indotineae* causes resistant dermatophytosis to terbinafine. The global spread of terbinafine-resistant *Trichophyton indotineae* strains with mutations in the squalene epoxidase gene is a major issue. This emerging species is now more frequently isolated in Europe and we report here two cases of *T. indotineae* tinea corporis in Switzerland, one with in vitro resistance to terbinafine and a second with in vitro susceptibility but a clinical resistance. Mycology isolation from cultures and sequencing ITS gene were used to confirm *T. indotineae* infection. In vitro antifungal susceptibility was tested in a microplate with a colorimetric detection of fungal viability for the determination of the minimal inhibitory concentration (MIC). Facing these emerging resistances and since there are a limited number of antifungal agents available to treat dermatophytosis, the early detection of terbinafine resistance should be a prerequisite in the management of *T. indotineae* infections.

## 1. Introduction

Dermatophytes are superficial keratinophilic fungi that cause major cutaneous infections with a global prevalence of 20–25% [1]. Trichophyton is the most frequent dermatophyte genus responsible for tinea corporis.

So far, tinea corporis successful cures remain dominated by terbinafine, an antifungal agent of the allylamine group, which is widely used per os and topically. Terbinafine inhibits squalene epoxidase thereby interfering with the biosynthesis of ergosterol, an essential cell membrane component [2]. Various point mutations in the squalene epoxidase (*SQLE*) gene which results in amino acid substitutions are found in terbinafine-resistant *Trichophyton* strains and are thought to convey resistance. However, among the different mutations in the *SQLE* gene reported so far, some are associated with in vitro and clinical terbinafine resistance whereas others are not. Since the mid-2010s resistant dermatophytosis has emerged as a major health problem in India and reached alarming proportions in this country, while expanding in many other countries around the world due to immigration, travel and then local transmission [3,4]. The availability of over-the-counter creams containing a steroid, antifungal, and antibacterial has likely favored such expansion [5].

Although the burden of this problem has not gained global attention, recalcitrant superficial dermatophytosis may become a potential global public health threat. Moreover, continued use of terbinafine in patients with recalcitrant dermatophytosis may favor the selection of resistant species responsible for the progression to chronic/recalcitrant infections. Most terbinafine-resistant strains belong to the newly described species *Trichophyton indotineae* [6,7].

The strains of this emerging species are now more frequently isolated in Switzerland and we report here two imported cases of *T. indotineae* tinea corporis, one showing terbinafine in vitro resistance and a second with in vitro susceptibility to terbinafine but with clinical resistance.

## 2. Cases


**Case 1**


A 32-year-old breastfeeding woman from India, living in Geneva since 2021, consulted for a pruritic erythemato-squamous dermatosis on the forearm. The skin lesion had started 3 years ago when she was living in India. The lesions had not regressed after 9 months of topical econazole treatment prescribed by a general practitioner in Spain. When she consulted in our dermatology clinic several slightly annular superficial erythematous and squamous plaques were found on both forearms (Figure 1).

Fluorescent microscopy examination of skin’s scales stained with blankophor revealed the presence of fungal septate filaments. A fungal isolate with morphological features corresponding to *T. mentagrophytes* was isolated in culture after seven days of incubation at 30 °C. However, ITS sequencing indicated that the strain was a *T. indotineae*. Sequencing of the SQLE gene revealed the presence of both Phe^397^Leu and A^448^Thre substitutions (Figure 2).

Antifungal susceptibility testing by the EUCAST method showed a high minimal inhibitory concentration (MIC) to terbinafine (>16 mg/L) and a low MIC to itraconazole (0.06 mg/L).

Due to breastfeeding, no oral therapy could be initiated and topical treatment with ketoconazole lotion was prescribed for 3 months instead. However, erythematous and inflammatory skin lesions remained and *T. indotineae* was still isolated by culture after three months (Figure 3). The patient was then lost to follow-up preventing any evaluation of the treatment outcome.


**Case 2**


A 26-year-old healthy man from Afghanistan, living in Switzerland since December 2021 consulted for a pruritic extensive dermatosis that occurred two months before his arrival. Clinically, skin lesions presented as erythematous annular plaques of the groins and multiple excoriated papules localized over the trunk, legs and arms (Figure 4).

Fungal filaments were found by direct microscopic examination of skin scales from inguinal folds. A *Trichophyton* species which was identified to be *T. indotineae* by ITS sequencing was isolated in culture after 12 days of incubation. DNA sequencing of the squalene epoxidase gene from the strain disclosed an Ala^448^Thr substitution (Figure 2). In vitro susceptibility testing by the EUCAST method showed a susceptible MIC to terbinafine at 0.06 mg/L.

A treatment combining oral terbinafine (250 mg/d) and topical ketoconazole was prescribed for 2 months. A partial improvement was obtained but erythematous and inflammatory skin lesions persisted and *T. indotineae* was still isolated two and three months after treatment initiation. To better understand this clinical resistance, we assessed terbinafine concentration in patient blood and found a value of 478 µg/L which was within the range for a well-monitored treatment, therefore, indicating good patient compliance. Terbinafine at the same dose and topical ketoconazole were maintained for one additional month and complete healing was achieved two months later. Thus a treatment duration of 4 months with oral terbinafine was required to cure this extensive tinea corporis (Figure 5).

## 3. Methods of Mycological Identifications and Susceptibility Testing

Specimens were collected in a sterile flask by scraping the edge of skin lesions with a sterile scalpel. Skin scales were then separated into two halves, one half for blankophor microscopy and one for culture. Cultures and strain identification was carried out as previously reported [8,9]. Briefly, skin samples were inoculated on Sabouraud chloramphenicol agar and Sabouraud chloramphenicol cycloheximide agar. Isolated strains were sub-cultured on Sabouraud dextrose agar for identification which was carried out by morphological examination of colonies and by MALDI-TOF [9]. As the identification was not conclusive by mass spectrometry, ITS sequencing was performed and DNA sequences were analyzed using nucleotide BLAST with type material sequences from the nucleotide collection (nr/nt).

In vitro analysis of antifungal susceptibility was tested in a microplate with a colorimetric detection of fungal viability based on metabolic activity. Microplate wells contain six antifungal molecules (amphotericin B, fluconazole, itraconazole, ketoconazole, griseofulvin and terbinafine) at various concentrations; ranging from 256 mg/L to 0.12 mg/L for fluconazole and from 16 mg/L to 0.008 mg/L for the others five molecules. Custom Sensititre^TM^ microplates were prepared by Thermo Scientific (Thermo Fisher Scientific Inc., Waltham, MA, USA) for amphotericin B, fluconazole, itraconazole, ketoconazole. For terbinafine and griseofulvin serial two-fold dilutions were prepared in DMSO and added to the well containing the fungal suspension. MIC determination was performed according to the EUCAST method [10] with minor modifications. Briefly, isolates were sub-cultured on Potato dextrose agar and incubated at 30 °C for 7 days. Harvested colonies were resuspended in 0.9% NaCl and 0.05% Tween 20. A microconidia suspension was prepared by filtration using filters with 11µm pore diameter and cell density adjusted to McFarland of 0.5. Two milliliters of the suspension were then diluted in 11 mL of Sensititre Yeast Susceptibility Inoculum Broth (Thermo Fischer Scientific Inc.) to obtain a theoretical final working solution of 4.10^5^ to 1.10^6^ CFU/mL. A hundred microliters were added to each well of the microplate. The viability and the counting of CFU were performed on Sabouraud dextrose agar according to EUCAST protocol. One hundred to 250 colonies were expected from an acceptable test suspension. A modification with the EUCAST method [10] was the reading of the cell viability by Alamar blue which changed from blue to pink when growth was present. The MIC value was defined by visual inspection of color change after 4–5 days of incubation at 30 °C. The first well with a blue color corresponded to the minimum inhibitory concentration for a given molecule. An antifungal plate illustrating a resistant strain to terbinafine (>16mg/L) was presented in Figure 6.

MIC values of the strains isolated from the two patients were shown in the Table 1.

Sequencing of the squalene epoxidase gene and the detection of mutations was performed as previously described by Hsieh and al [8].

## 4. Discussion

*T. mentagrophytes* internal transcriber space (ITS) genotype VIII was renamed *T. indotineae* and classified as a separate species in 2020. This species had already been detected between 2004 and 2013 in India, Australia, Iran and Oman. After 2014, a significant number of additional cases with recalcitrant dermatophytosis involving *T. indotineae* were reported in other countries and started to worry practitioners. The terbinafine resistance of this species was suspected because no clinical response to treatment and a worsening of skin lesions were observed in the patient despite a therapy that was supposed to be effective. Terbinafine-resistant dermatophytosis was first limited to India and its neighboring countries for a short period. During the last 3 years, terbinafine resistance was also reported in many European countries but was limited to patients coming from endemic regions or that have been traveling in endemic areas [11]. The highest numbers of cases in Europe were from Germany but several cases from France, Belgium and Denmark and now from Switzerland have been identified. A recent study reported that in India up to 76% of isolates display high MIC to terbinafine [12].

Dermatophytosis caused by *T. indotineae* often begins with tinea corporis, tinea cruris or tinea genitalis as inflammatory or hyperpigmented scaly and itchy lesions. *T. indotineae* is mainly transmitted from human to human instead of zoophilic transmission. Information such as the country of origin and travel history of the patient can help the clinician to suspect a resistant species of dermatophyte and carefully monitor the patient until complete healing.

The prolonged exposure to terbinafine of patients with recalcitrant dermatophytosis and with a sub-optimal treatment may favor species responsible for the progression to chronic/recalcitrant infections and the emergence of new mutations. Therefore, we believe that it is important to early identify dermatophyte species to seek for antifungal resistance when *T. indotineae* is isolated.

A recent study analyzing 498 isolates of *T. mentagrophytes/interdigitale* complex reported low MIC for itraconazole, miconazole, luliconazole, amorolfine, voriconazole, ketoconazole, and ciclopirox olamine compared to terbinafine, fluconazole and griseofulvin. As a low percentage of isolates had MIC above the upper limit of wild-type MIC (UL-WT) for itraconazole [13] the authors suggested that itraconazole could be considered the choice of first-line treatment.

The drug of choice for treating dermatophytosis caused by *T. indotineae* resistant to terbinafine is itraconazole. Expert Consensus on The Management of Dermatophytosis in India (ECTODERM India) recommends a dose of 100 mg twice daily for 2 to 4 weeks in naïve cases and more than 4 weeks with a doubling dose in recalcitrant cases [14].

The Indian Association of Dermatologists, Venereologists and Leprologists (IADVL) also developed a task force against recalcitrant tinea (ITART). They suggested using a combination of systemic and topical antifungals for the treatment of glabrous tinea that should be maintained for 2 weeks after clinical resolution. The use of topical corticosteroids should be strictly avoided, even in combination with antifungals [15].

For our first patient (case 1), we reported a clinical resistance involving a *T. indotineae* strain with high terbinafine MIC and two mutations Ala448Thr and Phe397Leu in the squalene epoxidase gene with the latter known to confer this resistance. The choice of treatment was limited as the patient was in late pregnancy and then breastfeeding. To date, no information is available on the clinical usage of itraconazole during breastfeeding but there are limited data that suggest that during breastfeeding, maternal itraconazole results in less than the 5 mg/kg daily doses that have been recommended to treat infants. However, until more data become available, an alternate drug may be preferred.

For our second patient (case 2), the lesions persisted for a long time and more than 3 months of oral terbinafine was needed to eradicate the infection although the isolated *T. indotineae* strain had low terbinafine MIC. An explanation for this phenomenon is the “environmental pressure” from *T. indotineae* and emphasize the importance of textile disinfection to reduce transmission and reinfection rate. Indeed, since *T. indotineae* dermatophytosis is often hard to treat, it is challenging to differentiate the true terbinafine resistance of the strain from a *de novo* contamination occurring by close contact with infected people living with the patient, the persistence of a potential reservoir such as a pet or questionable hygiene. The other key issue for the management of dermatomycoses is advice on patient environmental decontamination which plays an essential role to avoid reinfection, especially for recalcitrant/chronic dermatophytosis due to *T. indotineae*. A study already published in 1984 showed that *T. rubrum* survived for <12 weeks on a towel while *T. mentagrophytes* can persist for more than 25 weeks [16]. This can also explain why *T. indotineae* is so hard to eradicate.

Facing these emerging resistances and since there is a limited number of antifungals available to treat dermatophytosis, clinicians need to be better guided to choose the appropriate treatment. In *T. indotineae* dermatophytosis, antifungal susceptibility testing should become the standard in laboratories to monitor dermatophyte resistance in the community. However, the microdilution method is time-consuming and very tricky to perform. The colorimetric detection of cell growth helps to visualize CMI but standardization of the inoculum to obtain reproducible values is a real challenge. Fortunately, *T. indotineae* isolates usually sporulate well and their antifungal susceptibility test is relatively easy to read after 4 to 5 days. For dermatophytes that sporulate poorly, screening on a plate containing 0.2 mg/L of terbinafine is a good alternative for early detection of a probable resistance but it needs to be confirmed by susceptibility testing.

It should be also emphasized that MIC values do not always correlate with clinical response to antifungal drugs and that clinical breakpoints have yet to be established for dermatophytes. However, EUCAST elaborated epidemiological cut-off values (ECOFFs) to define if a strain is rather a wild-type strain or not. ECOFF is defined as the highest MIC of isolates that are not known to have resistance and are, therefore, considered representative of the wild type. Recently, *T. indotineae* ECOFFs were established from a shared isolate collection tested in 10 laboratories [17], with a value of 0.125 mg/L for terbinafine and 0.25 for itraconazole mg/L. For *T. rubrum*, the most frequent species isolated in dermatological laboratories from our country, wild-type ECOFFs are different from those of *T. indotineae*. Based on several recent studies it is now well admitted that a terbinafine MIC > 2 mg/L means the resistance of the strain not only for *T. indotineae* but also for other dermatophytes species.

Sequence analysis of the *SQLE* gene is also useful to establish a correlation between mutations and MIC and to monitor the emergence and spread of resistance in dermatophytes. In a study of 52 *T. indotineae* isolates from India, missense mutations in the *SQLE* gene leading to different amino acid substitutions were analyzed and different terbinafine MIC were determined [18]. Sixty-five percent of the strains harbor point mutations with amino substitutions Phe^397^Leu or Phe^397^Leu and Ala^448^Thr and had terbinafine MIC > 16 mg/L. For 4% of the isolates, the MIC was 0.5 mg/L, a value above the ECOFF and with amino acid substitution Phe^415^Val or Leu^393^Ser. For the other strains with terbinafine MIC < 0.25mg/L, the two amino acid substitutions His^440^Tyr or Ala^448^Thr were detected. In these data, *T. indotineae* showed an almost bimodal distribution with a majority of strains with in vitro resistance.

The two cases reported here showed the same characteristics, one with a low terbinafine MIC and the other with a high MIC.

In conclusion, in this epidemiological context, we believe that monthly clinical monitoring and antifungal susceptibility tests should now be proposed more systematically in dermatophytic infections that do not respond to treatment. Larger information on antifungal stewardship is encouraged among practitioners but also biologists.

## Figures and Tables

**Figure 1 jof-09-00733-f001:**
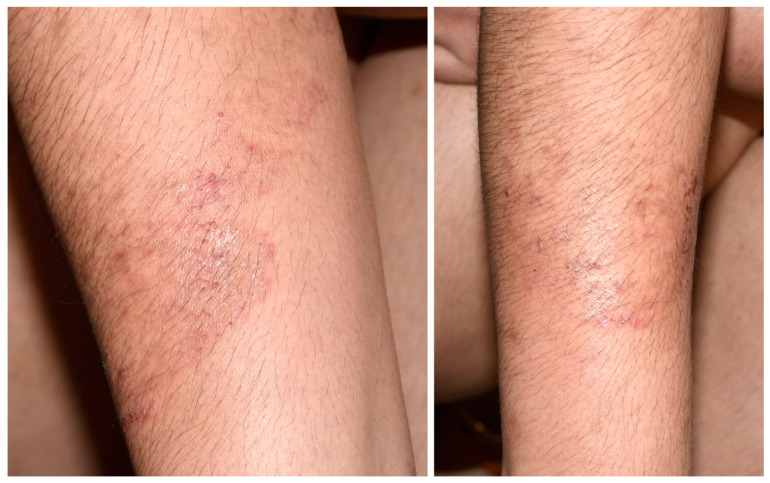
Erythematous annular lesions with a scaly raised border and central clearing.

**Figure 2 jof-09-00733-f002:**
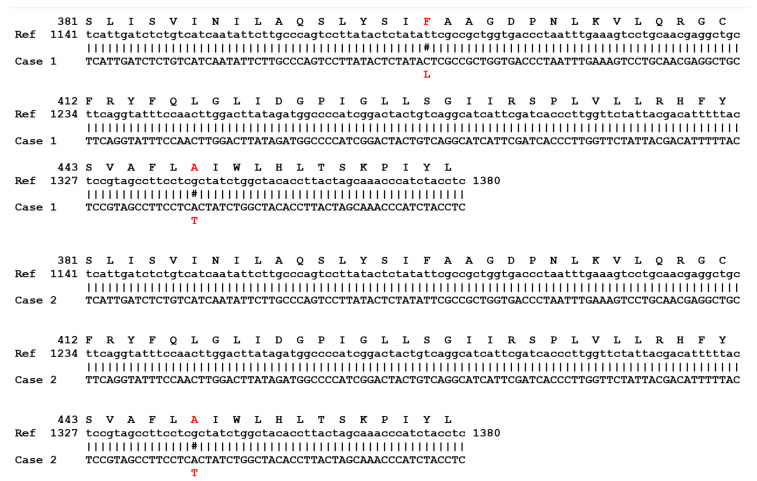
Sequence alignments of squalene epoxidase C-terminal showing that case 1 (Genebank accession number OR047662) contains both Phe397Leu and Ala448Thr substitutions while case 2 (Genebank accession number OR047661) only harbors the Ala448Thr. Alignments were performed using serial cloner freeware with KU242352 ORF sequence from *T. mentagrophytes* as reference). # denoted substitutions. Residues and nucleotides are numbered from the first coding codons and residues of SQLE ORF Case 1, Case 2.

**Figure 3 jof-09-00733-f003:**
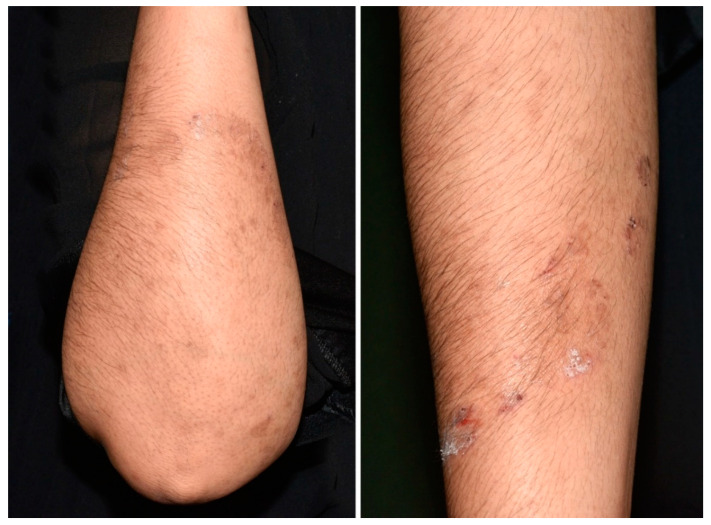
Classic annular lesions with a border composed of some slightly scaly papules.

**Figure 4 jof-09-00733-f004:**
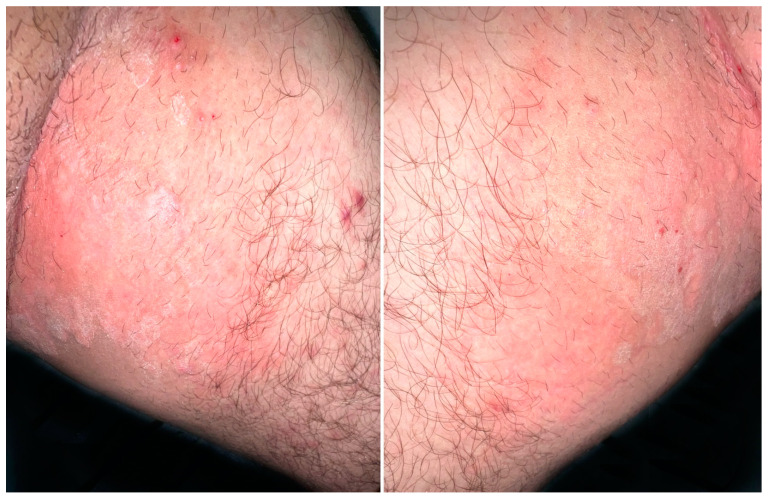
Thin, erythematous plaques with an arciform squamous border on the upper inner thighs.

**Figure 5 jof-09-00733-f005:**
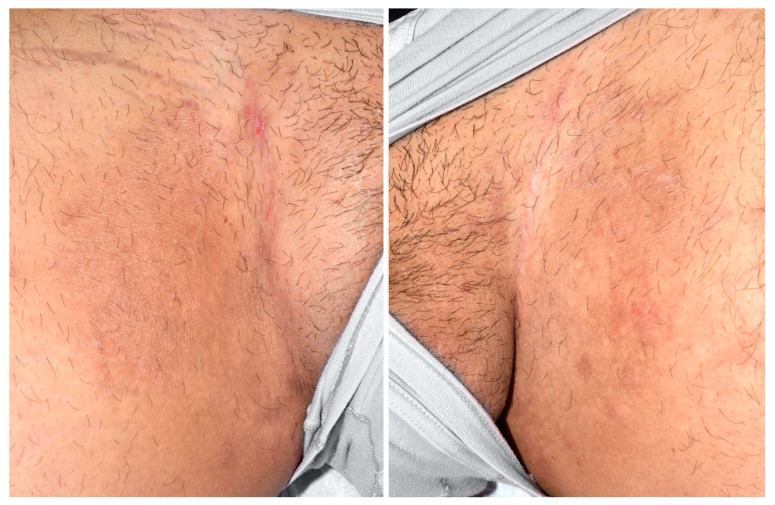
Complete healing of skin lesions.

**Figure 6 jof-09-00733-f006:**
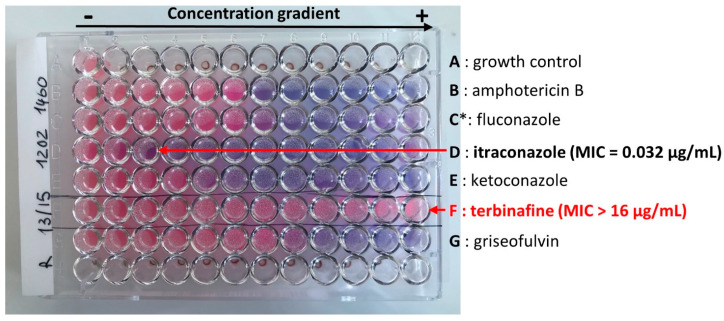
*Trichophyton indotineae*, terbinafine resistant strain. Sensititre^TM^ microplate with various antifungal compounds. A: growth control, B: amphotericin B, C: fluconazole, D: itraconazole, E: ketoconazole, F: terbinafine, G: griseofulvin. Rows A, B, D, E, F: increasing concentration from 0.008 µg/mL to 16 ug/mL. Row C*: increasing concentration from 0.12 µg/mL to 256 µg/mL.

**Table 1 jof-09-00733-t001:** MIC values of *Trichophyton indotineae* against six drugs ^a^.

	Value (mg/L)					
Species and Drugs	AMPHO B	FCZ	ITZ	KTZ	TBF	GRI
*T. indotineae* (case 1)	1	32	0.06	0.5	>16	2
*T. indotineae* (case 2)	0.5	64	0.25	1	0.06	1

^a^: AMPHO B, amphotericin B; FCZ, fluconazole; ITZ, itraconazole; KTZ, ketoconazole; TBF, terbinafine; GRI, griseofulvin.

## Data Availability

The *Trichophyton indotineae* strain 37516497 squalene epoxidase sequence (case 1) can be found under the number OR047662 in the Genbank database. The *Trichophyton indotineae* strain 37346462 squalene epoxidase sequence (case 2) can be found under the number OR047661.

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
