# Peer review of "Towards an Early Clinical and Biological Resistance Detection in Dermatophytosis: About 2 Cases of Trichophyton indotineae"

_jof, 2023, doi:10.3390/jof9070733_

Round 1

Reviewer 1 Report

Dear Authors,

The cases presented are very interesting in relevant int he field of dermatomycosis.

Please check these minor things:

1) In line 120 add a point between ...: growth is present. The antifungal...

2) In figure 5: please organize the labels: some legend are cutted (incomplete). Also I don´t think that is necessary to use the word row so many times).

3) In line 155: to write T.indotineae in italics: T.indotineae 

Author Response

Dear Authors,

The cases presented are very interesting in relevant in the field of dermatomycosis.

Please check these minor things:

1) In line 120 add a point between ...: growth is present. The antifungal.:

  • correction made

2) In figure 5: please organize the labels: some legend are cutted (incomplete). Also I don´t think that is necessary to use the word row so many times).  

  • The figure 5 was partially shifted when the text was edited by the editor. It has been worked and the repetition of the word arrow has been removed.

3) In line 155: to write T.indotineae in italics: indotineae

  • correction made

Reviewer 2 Report

The authors in this manuscript found that two cases of T. indotineae tinea corporis in Switzerland which show terbinafine in vitro resistance, and in vitro susceptibility but a clinical resistance. They isolated the strains, and identified the features of strains. The results are interesting but preliminary I have several concerns which needs to be respond by the authors as follows:  

1. The authors should provide the DNA sequencing or alignment results in the manuscript.

2. The MIC results should provide as table in the manuscript.

3. The discussion is over-stated.

4. the methods section should described in detail.

Author Response

The authors in this manuscript found that two cases of T. indotineae tinea corporis in Switzerland which show terbinafine in vitro resistance, and in vitro susceptibility but a clinical resistance. They isolated the strains, and identified the features of strains. The results are interesting but preliminary I have several concerns which needs to be respond by the authors as follows:  

  1. The authors should provide the DNA sequencing or alignment results in the manuscript.
  • The alignment results of the two squalene epoxidase genes have been added and shown in Figure 2. They have been deposited to Genbank with accession number reported in figure 2 legend.

  1. The MIC results should provide as table in the manuscript.
  • The table has been added in the text (row 166)

  1. The discussion is over-stated.
  • We appreciate this comment concerning the discussion. Out of these 2 cases, we are following new cases in our outpatient clinic showing same clinical lesions and resistance profile. We are convinced that indotineae infections present/become a specific concern for dermatologists, biologists and general practitioners. Moreover, we believe that the global spreading of resistant dermatophytes requires an early species detection especially in patients who has travelled to or come from an endemic area. Supporting our statement, a recent review from the American Academy of dermatology have just been published https://www.aad.org/dw/dw-insights-and-inquiries/archive/2023/tinea-gone-wild-trichophyton-indotineae.

  1. the methods section should described in detail.

  • The method has been described in more detail (row 123 to 146)

Reviewer 3 Report

Manuscript “Towards an early clinical … Trichophyton indotineae” by Giuseppe Russo et al presents two cases of tinea corporis with Trichophyton indotineae from Switzerland.             

Were results read manually?

Please state explicitly the inoculum used.

Please add some history data for the second patient, i.e. immunity status and if he was taking the pills properly.

It remains somewhat unclear if the two cases were imported (probably) or acquired locally. Please clarify this point.

Please deposit the sequences to a public database such as GenBank and add the accessions to the manuscript.

Fig. 5. Please add some arrows showing the wells corresponding to the MICs, for the benefit of readers inexperienced with the method.

Please add references for use of MALDI-TOF for identification of the species, and for detection in Switzerland.

Minor mistakes

Line 145; please change “T.indotineae” to “T. indotineae”. Also change all taxon names in similar way.

Please italicize all taxon names in the text and references.

Author Response

Manuscript “Towards an early clinical … Trichophyton indotineae” by Giuseppe Russo et al presents two cases of tinea corporis with Trichophyton indotineae from Switzerland.             

Were results read manually?

  • Two sentences have been added to the manuscript to explain how the MIC results were read. The reading was done manually by detecting the change in color from blue to pink indicating a growth in the well. (row 141 to 146)

Please state explicitly the inoculum used.

  • The description of how the inoculum was prepared and its theoretical value have been added (row 132 to 141)

Please add some history data for the second patient, i.e. immunity status and if he was taking the pills properly.

  • Our apologies for the We added “healthy” to the sentence “A 26-years-old healthy man from Afghanistan” (row 87). For the medication compliance, we mentioned the assessment of a terbinafine plasmatic level that found a normal value; thus, indirectly indicates a good compliance and that the patient took the pills properly. To better clarify this point we added “ this confirms a good patient compliance” to this sentence: “To better understand this clinical resistance, we assessed terbinafine concentration in patient blood and found a value of 478 ug/L which was within the range for a well-monitored treatment, therefore indicating a good patient” (rows 103-105).

It remains somewhat unclear if the two cases were imported (probably) or acquired locally. Please clarify this point.

  • We appreciate this comment and fully agree with the reviewer that this point was not well clarified in the text. We added “imported” to the sentence “The strains of this emerging species are now more frequently isolated in Switzerland, and we report here two imported cases of indotineae tinea corporis” (rows 46-49). For case n1 we mentioned that the patient already presented skin lesions when she lived in India : “Skin lesion had started 3 years ago, when she was living in India” (rows 50-51). For case n2 we clarified the onset of lesions “A 26-years-old healthy man from Afghanistan, living in Switzerland since December 2021 consulted for a pruritic extensive dermatosis that occurred two months before his arrival.” (rows 87-89).

Please deposit the sequences to a public database such as GenBank and add the accessions to the manuscript.

  • The sequences have been deposited to Genbank and the accession numbers are reported in figure 2 legend. The alignment results of the two squalene epoxidase genes have been added and shown in Figure 2.

Fig. 5. Please add some arrows showing the wells corresponding to the MICs, for the benefit of readers inexperienced with the method.

  • An arrow was added for terbinafine (>16ug/mL) and for itraconazole (MIC=0.032ug/mL)

Please add references for use of MALDI-TOF for identification of the species, and for detection in Switzerland.

  • The reference was already in the manuscript: reference N°9

Minor mistakes

Line 145; please change “T.indotineae” to “T. indotineae”. Also change all taxon names in similar way.

Please italicize all taxon names in the text and references.

  • Correction made

Round 2

Reviewer 2 Report

All my concerns have been addressed!